# Mulching Practices Improve Soil Moisture and Enzyme Activity in Drylands, Increasing Potato Yield

Wenhuan Song [1,2], Fanxiang Han [3], Zhengyu Bao [1,2], Yuwei Chai [1,2], Linlin Wang [1,2], Caixia Huang [4], Hongbo Cheng [5] and Lei Chang [1,2,*]

1 State Key Laboratory of Aridland Crop Science, Gansu Agricultural University, Lanzhou 730070, China; songwh@st.gsau.edu.cn (W.S.); baozy@gsau.edu.cn (Z.B.); chaiyw@gsau.edu.cn (Y.C.); wangll@gasu.edu.cn (L.W.)
2 College of Agronomy, Gansu Agricultural University, Lanzhou 730070, China
3 College of Geography and Environmental Engineering, Lanzhou City University, Lanzhou 730070, China; hanfx@lzcu.edu.cn
4 College of Water Conservancy and Hydropower Engineering, Gansu Agricultural University, Lanzhou 730070, China; huangcx@gsau.edu.cn
5 College of Life Science and Technology, Gansu Agricultural University, Lanzhou 730070, China; chenghb@gsau.edu.cn
* Correspondence: changlei@gsau.edu.cn; Tel.: +86-139-1988-3433

**Abstract:** Mulch is an important measure for improving agricultural productivity in many semiarid regions of the world. However, the impacts of various mulching materials on soil hydrothermal characteristics, enzyme activity, and potato yield in fields have not been comprehensively explored. Thus, a two-growing-season field experiment (2020–2021) with four treatments (SSM, straw strip mulching; PMP, plastic film mulching with large ridge; PMF, double ridge-furrow with full film mulching; and CK, no mulching with conventional planting as the control) was conducted to analyze soil hydrothermal and soil enzyme activities and potato yield on the semiarid Loess Plateau of Northwest China. The results indicated that mulching practices had a positive effect on the soil moisture, with SSM, PMP, and PMF increasing by 7.3%, 9.2%, and 9.2%, respectively, compared to CK. Plastic film mulching significantly increased the soil temperature by 1.3 °C, and straw mulching reduced the soil temperature by 0.7 °C in the 0–30 cm soil layers of the whole growth period. On average, SSM, PMP, and PMF increased soil urease activity in 0–40 cm soil layers by 14.2%, 2.8%, and 2.7%, respectively, and enhanced soil sucrase activity by 19.2%, 8.6%, and 5.7%, respectively, compared with CK. Plastic film mulching increased soil catalase activity by 9.6%, while SSM decreased by 10.1%. Mulching treatments significantly increased tuber yield and water use efficiency based on dry tuber yield (WUE), and SSM, PMP, and PMF increased tuber yield by 18.6%, 31.9%, and 29.7%, enhanced WUE by 50%, 50%, and 57.0% over CK. The correlation analysis revealed that soil moisture was the main factor influencing tuber yield (r = 0.95**). Mulching could improve the soil hydrothermal environment, regulate soil enzyme activities, and promote yield increase. As a sustainable protective mulching measure, straw strip mulching is conducive to improving the ecological environment of farmland and the sustainable development of regional organic agriculture.

**Keywords:** dryland farming; mulching; soil moisture; soil enzyme activities; tuber yield

## 1. Introduction

Drought and water scarcity are major agricultural challenges; population growth and climate change have further exacerbated water scarcity, threatening food security worldwide [1]. Dryland agriculture makes up 33% of the arable land in China, with 56% of it located in the northwest region [2]. Potato (*Solanum tuberosum* L.) is the world's first major non-cereal food crops, and the area of potato cultivation in the semiarid rainfed agricultural region in Northwest China was $1.3 \times 10^6$ ha$^{-1}$, accounting for 36% of the total area under

potato cultivation in China [3]. The mismatch between natural rainfall and crop moisture demand and supply, resulting in low and unstable crop yields, is the main limiting reason for the development of the potato industry [4]. Therefore, how to effectively use natural precipitation during the potato growth period, reduce ineffective evaporation, optimize soil moisture and temperature, and improve crop root microenvironment to increase crop yield are critical issues in the efficient use of moisture and temperature in dryland agriculture.

Soil surface mulching is an important way to conserve soil moisture and raise temperatures in rainfed agroecosystems, which could increase crop yields and moisture productivity [5]. Mulching with plastic film and straw could enhance rainfall infiltration, reduce soil evaporation [6], regulate soil temperature [7], improve soil structure, and enhance the hydrothermal environment suitable for crop growth. Ref. [8] demonstrates that the average yield of wheat, maize, and potato can increase by 24.3%, and water use efficiency can increase by 27.6%. However, as plastic film usage increases, the accumulation of plastic residues in the soil will change the soil physical and hydraulic properties, thus causing a decline in the quality of cultivated land [9]. Straw mulching is a straightforward and user-friendly method, which is more consistent with green agriculture, although conventional full straw mulching practice provides good moisture retention. In cold regions, full straw mulching practice will delay crop emergence, and inhibit early growth and development; this often comes with the risk of reduced yield [10]. Straw strip mulching technology divides the plot into planting strip and mulching strip; the two strips are arranged alternately, the whole corn stalks are mulched in the mulching strip, and potatoes are sown in the planting strip. Considering the constraints of singular full straw mulching or plastic film mulching technology, straw strip mulching integrates the distinct benefits of plastic film and straw mulching. While reducing the use of plastic film, it could achieve the purpose of restraining evaporation and preserving soil moisture, harvesting rainwater, increasing infiltration, and improving crop yield [11]. It has been popularized and applied in wheat and potato production [3,7].

As biocatalysts in soil, soil enzymes engage in the metabolism and transformation of soil nutrients and optimize physical and chemical properties of soil [12]. Soil enzyme, as an important biological indicator for assessing soil conditions, plays a vital role in soil transformation. Urease in soil can break down urea into ammonia as part of the nitrogen cycle, supplying necessary nitrogen for crop growth [13]. Catalase, as an important enzyme in microbial metabolism, can catalyze the decomposition of hydrogen peroxide to release oxygen and reduce toxic effects of hydrogen peroxide on soil [14]. Sucrase is an important medium of carbohydrates in plants and promotes the hydrolysis of sucrose [15]. Soil hydrothermal properties are an important factor affecting soil enzyme activity; these physical and chemical indexes can further reflect soil quality. Mulching changes soil moisture and temperature status, which will inevitably affect the soil enzyme activity. Previous studies have found a strong positive effect of moisture on soil enzyme activity, with higher moisture content leading to better binding of soil enzymes to substrates, which in turn promotes root development and overall plant growth [16]. It could also influence environmental factors such as soil temperature and moisture, which could affect crop uptake and utilization [17]. Seo et al. [18] concluded that elevated temperatures enhance phenol oxidase activity and increase the reaction rate of phenol oxidase to promote soil carbon cycling. Yang et al. [19]. pointed out that at the tuber swelling stage, sucrase and catalase activities of plastic film mulch were significantly higher than of straw mulch, with sucrase activities increasing by 18.71% and catalase activities increasing by 17.44%; the rate of large potatoes and yield were increased by 20.15% and 17.90% more than CK. Currently, the yield-increasing mechanism of straw strip mulching technology is mainly focused on soil hydrothermal and agronomic indexes, although the potential of soil hydrothermal and soil enzyme activity changes on potato yield enhancement has not been fully explored. The effects of mulching practices on soil hydrothermal and enzyme activities along with tuber yield should be explored, to improve the relationship between hydrothermal and nutrient requirements for potato growth and agronomic practices. Studies have found that

the enhancement of soil hydrothermal conditions, resulting from the use of plastic film mulch and straw strip mulch, plays a vital role in increasing potato yields. Does the soil hydrothermal environment affect soil enzyme activity? How do mulching practices impact soil hydrothermal conditions, enzyme activity, and tuber yields? Therefore, the purpose of this study is as follows: (1) to analyze and compare the effects of different mulch treatments on potato soil hydrothermal conditions, soil enzyme activities, and potato yield; (2) to clarify the mechanism of soil hydrothermal effects on enzyme activity differences under different mulch treatments; and (3) to assess the application value of various mulching techniques for potato cultivation in the semiarid rainfed farming system.

## 2. Materials and Methods

### 2.1. Survey of the Test Area

The experiment was conducted at Tongwei Modern Dryland Circular Farming Experiment Station, Gansu Province, China (35°11′ N, 105°19′ E; altitude 1740 m). Situated in the heart of Gansu Province, at an elevation of 1750 m above sea level, this region experiences an average annual evaporation rate of 1500 mm, average yearly precipitation of 390.6 mm (ranging from 250 to 550 mm), and an average annual temperature of 7.2 °C. This area is a classic example of a rain-dependent agricultural area on the Loess Plateau, where crops are harvested once annually. The experimental site's soil is classified as loessal soil. The plow layer (0–30 cm) has a soil bulk density of 1.25 g·cm$^{-3}$. The available nitrogen, phosphorus, and potassium contents are 0.8 g·kg$^{-1}$, 0.01 g·kg$^{-1}$, and 0.12 g·kg$^{-1}$, respectively. The distribution of precipitation and air temperature during the experimental period is shown in Figure 1. The precipitation of potato growth periods were384.1 and 278.7 mm in 2020 and 2021. The classification of the experimental year was based on the total precipitation during the entire growing season of potatoes and the drying index (DI). A wet year was defined as DI > 0.35, a normal year as $-0.35 \leq$ DI $\leq 0.35$, and a drought year as DI < $-0.35$. In 2020, the DI was 1.1, and in 2021, it was $-0.2$. As a result, 2020 was categorized as a wet year and 2021 was classified as a normal year.

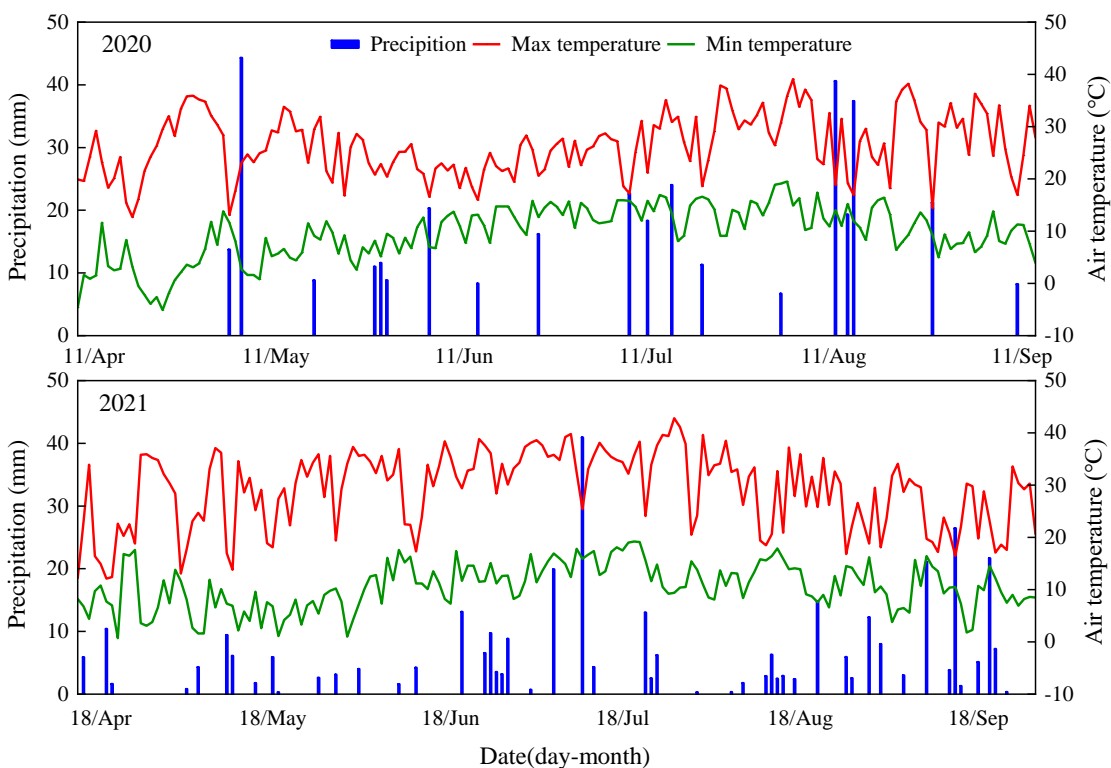

**Figure 1.** Precipitation and daily air temperature during the potato growth period in experimental years.

### 2.2. Experimental Design and Field Management

The tested material is "Longshu 7", which belongs to the medium–late maturing variety. Random block arrangement was used in the experiment. Four treatments were applied in all experiments: straw strip mulching (SSM), plastic film mulching with large ridge (PMP), double ridge-furrow with full film mulching (PMF), and no mulching with conventional planting (CK, the control) (Figure 2). Three replicates were performed for each treatment using a randomized block design, with a plot area of 80 square meters (16 m × 5 m). SSM: The 60 cm of straw mulch belt and planting belt were arranged alternately. The straw mulch belt was covered with whole maize stalk, the mulching amount was about $5.55 \times 10^4$ plants ha$^{-1}$ with a sowing depth of 0.1 m, and the row spacing was 60 cm. PMP: Ridge width of 70 cm, height of 15 cm, and ridge width of 50 cm. The mulch was ridged with the black plastic film of width 90 cm, the ridge was not covered, and 2 rows were sowed in each ridge, with the row spacing being 60 cm. PMF: Large ridge width 70 cm, high for 20 cm, small ridge width 40 cm, high for 15 cm; water seepage bandwidth for 10 cm; the whole ground was mulched with black film with width of 120 cm. Two rows per ridge, row spacing 60 cm. CK: Flat cropping without mulch, planting with equal row spacing when sowing, interrow distance 60 cm. The distance between plants was 33 cm.

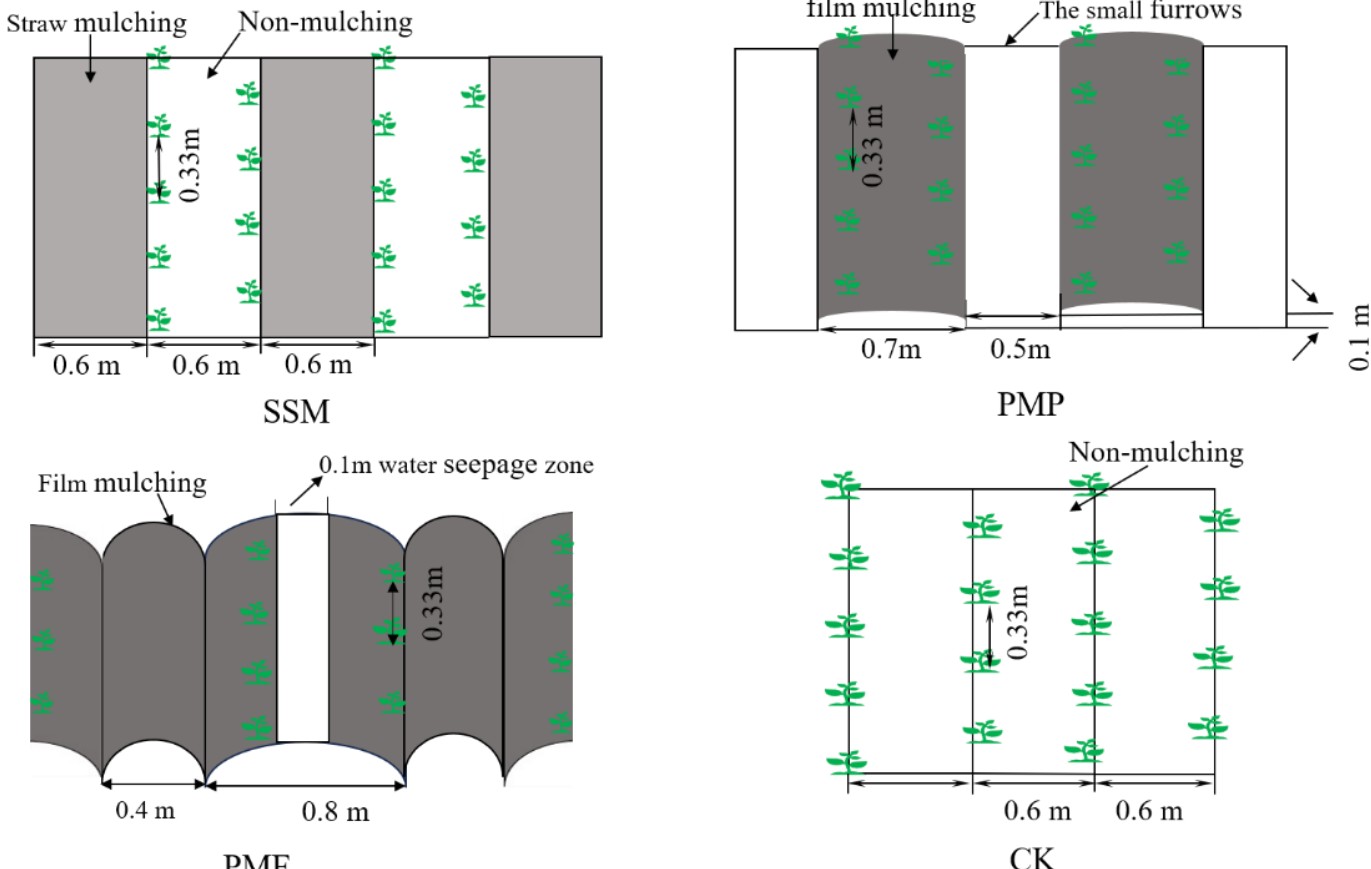

**Figure 2.** Schematic diagram of the experimental treatment. SSM, straw strip mulching; PMP, plastic film mulching with large ridge; PMF, double ridge-furrow with full film mulching; CK, no mulching with conventional planting.

The test site was planted with wheat as the previous crop, and the test site was deep tilled once and rotary plowed twice before mulching, then mulched and covered with straw. All fertilizers (pure N 120 kg·ha$^{-1}$, pure P$_2$O$_5$ 90 kg·ha$^{-1}$) were evenly applied as basal dressings. Potatoes were sown on April 15–18 and harvested in October. Throughout the growth period, no fertilizer was used on the potatoes, and the field was managed

consistently across all treatments. The planting density was $5.25 \times 10^4$ plants $ha^{-1}$, with a planting depth of 0.1 m.

### 2.3. Determination Items and Methods

2.3.1. Soil Moisture Content

Soil drills were used to sample potatoes among planting rows at sowing, seedling, budding, tuber formation, tuber expansion, starch accumulation, and maturity stages. In soil samples taken from 8 soil layers—0–20, 20–40, 40–60, 60–90, 90–120, 120–150, 150–180, and 180–200 cm (M1)—the soil water content was determined by means of the drying method, where soil samples were dried at 105 °C until reaching a constant weight (M2). Subsequently, the soil water content (SWC) and soil water storage (SWS) were calculated based on the collected data.

$$SWC\ (\%) = (M1 - M2)/M2 \times 100\% \tag{1}$$

$$SWS\ (mm) = h \times \rho \times \omega \times 10 \tag{2}$$

where h is soil depth (cm); $\rho$ is soil volume mass (g $cm^{-3}$); and $\omega$ is soil mass water content (%).

2.3.2. Soil Temperature

Soil temperature was measured by an iButton temperature recorder, which was placed in the planting zone and mulch zone, respectively. It was divided into 6 soil layers at 5, 10, 15, 20, 25, and 30 cm, and the recording interval was set to 1 h; the measurement period coincides with soil moisture.

2.3.3. Soil Enzyme Activity

At seedling, tuber bulking, and maturity stages, samples were taken with a soil auger from 0 to 40 cm, one layer of soil per 10 cm. Allowed to dry naturally, soil enzyme activities were determined by 1 mm sieve. Soil urease activity was determined by the sodium phenol–sodium hypochlorite colorimetric method and expressed as milligrams NH3-N in 1 g soil after 24 h. Catalase activity was determined by potassium permanganate titration, expressed as milligrams of 0.02 mol/L potassium permanganate, and consumed by titration of 1 g of soil after 20 min. Sucrase activity was determined using 3.5-dinitrosalicylic acid to calculate the quality of glucose produced in 1 g of dry soil after 24 h [20].

2.3.4. Tuber Yield

Following harvest in every plot, fifteen plants were chosen at random for indoor examination and were sorted into three categories according to their fresh weight: large potato (>150 g), medium potato (75 g), and small potato (<75 g). The number of potatoes in each grade was counted and weighed, and the commercial potato rate was calculated. Commodity rate (%) = ($\geq$75 g of tuber weight/output of tuber) $\times$ 100%.

### 2.4. Statistical Analysis

Microsoft Excel 2023 and origin 2022 (Origin Lab, Northampton, MA, USA) were used for data collation and mapping, SPSS (version 24.0, IBM SPSS Inc., Chicago, IL, USA). The least significant difference method was used to determine mean differences between treatments (LSD) at $p \leq 0.05$.

## 3. Results

### 3.1. Soil Moisture

3.1.1. Soil Moisture during the Whole Growth Period

Mulching cultivation practices could significantly increase soil moisture storage of all soil layers throughout the growth season. The average soil moisture in the upper layer (0–60 cm), middle layer (60–120 cm), and lower layer (120–200 cm) increased by 8.8%, 9.2%, and 7.6%, respectively. The increase was PMP > PMF > SSM (Figure 3). Plastic film and

straw strip mulching increased soil moisture of the upper layer by 6.8% and 5.8%, soil moisture of the middle layer by 10.0% and 9.6%, and soil moisture of the lower layer by 9.7% and 5.5% compared with CK in the wet year. In particular, PMP and PMF increased soil moisture of the upper layer by 7.4% and 6.2%, soil moisture of the middle layer by 9.7% and 10.3%, and soil moisture of the lower layer by 8.5% and 10.9% over the CK treatment, respectively. Plastic film mulching increased the soil moisture of upper, middle, and lower layers by 13.4%, 8.8%, 7.3%, respectively. Straw strip mulching improved by 12.8%, 8.6%, and 6.8% in the normal year, among which PMP and PMF treatments increased by 13.9%, 8.9%, and 7.9%, and 12.8%, 8.6%, and 6.8%, respectively, compared to CK. Compared to SSM, plastic film mulching resulted in significantly higher soil moisture in the upper and lower layers, by 33.5% and 48.7%, respectively. However, there was no significant variance in the middle soil moisture when compared to the SSM treatment.

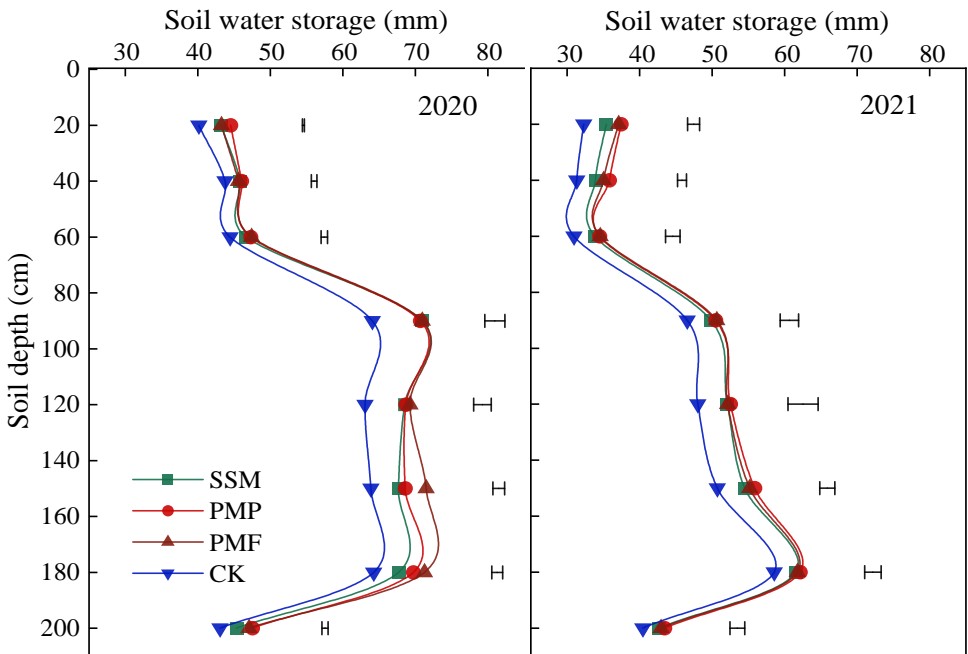

**Figure 3.** Changes in soil moisture profile in the whole growth period during 2020–2021. SSM, straw strip mulching; PMP, plastic film mulching with large ridge; PMF, double ridge-furrow with full film mulching; CK, no mulch with conventional planting. Bar indicates LSD = 0.05. The same as below.

3.1.2. Average Soil Moisture in 0–200 cm Layer at Different Growth Stages

Mulching increased average soil moisture in the 0–200 cm layer of potato from sowing to maturity by 8.4%, and the increase was manifested as PMF > PMP > SSM (Figure 4). Plastic film mulching and straw strip mulching respectively increased the average soil moisture of potato at various growth stages by 6.8% (4.1–10.7%) and 8.7% (6.1–12.4%) in the wet year. Among them, PMP and PMF increased by 8.2% and 9.2% over the CK treatment, respectively. SSM and PMP had the largest increase at starch accumulation period, and PMF increased the largest at seedling stage. Compared with mulching treatments, average soil moisture with SSM during the period from sowing to the tuber bulking stage was lower by 3.2% than that of plastic film mulch, while that of SSM at maturity stage was increased by 2.9%. In the normal year, plastic film and straw strip mulching increased average soil moisture at each growth stage by 9.8% (6.1–13.8%) and 7.0% (4.6–9.2%), compared with CK, respectively. Among them, PMP and PMF increased by 10.3% and 9.3%, respectively, over CK. SSM practice had the greatest increase at budding and plastic film mulching had the greatest increase at seedling. Compared with straw strip mulching, plastic film mulching increased soil moisture by 2.0% from sowing to maturity stage.

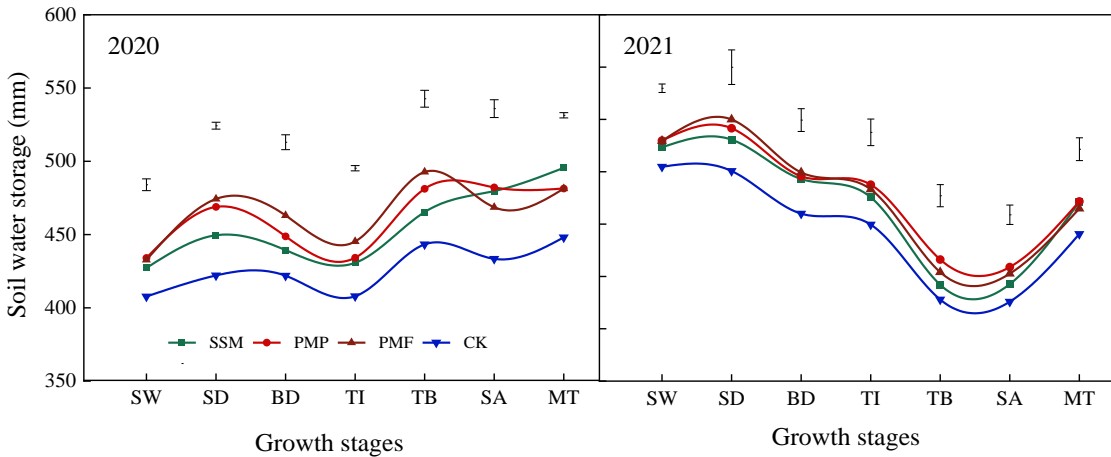

**Figure 4.** Dynamics of 0–200 cm soil moisture during different growth stages in 2020–2021. SW: sowing stage; SD: seedling stage; BD: budding stage; TI: tuber initiation stage; TB: tuber bulking stage; SA: starch accumulation stage; MT: maturity.

*3.2. Soil Temperature*

3.2.1. Soil Temperature of 0–30 cm Layer during the Whole Growth Period

The soil temperature of mulching practices during the whole growth period of potato decreased with the increase in soil depth. Straw strip mulching showed a cooling effect in 0–30 cm, while plastic film mulching showed a warming effect (Figure 5). In the wet year, the average soil temperature of plastic film mulching significantly increased 1.3 °C (1.0–1.7 °C) in 0–30 cm layer; the increase was PMF > PMP. Compared with CK, PMP significantly increased the temperature by 1.3 °C and PMF significantly increased the temperature by 1.5 °C in all soil layers, straw strip significantly decreased the temperature by 0.6 °C (0.4–0.8 °C) on average in all soil layers, and the greatest variation was found in the 10 cm layer in all mulching treatments. In the normal year, plastic film mulching increased the soil temperature of potato in 0–30 cm by 1.2 °C (1.0–1.3 °C); the increase showed that PMP>PMF. PMP and PMF increased the temperature by 1.2 °C and 1.1 °C compared with CK on average in each soil layer, and both of them increased mostly in the 30 cm layer. The average decrease in SSM in each soil layer was 0.8 °C (0.5–1.0 °C) lower than that of CK, and the decrease was largest in 20 cm soil layer.

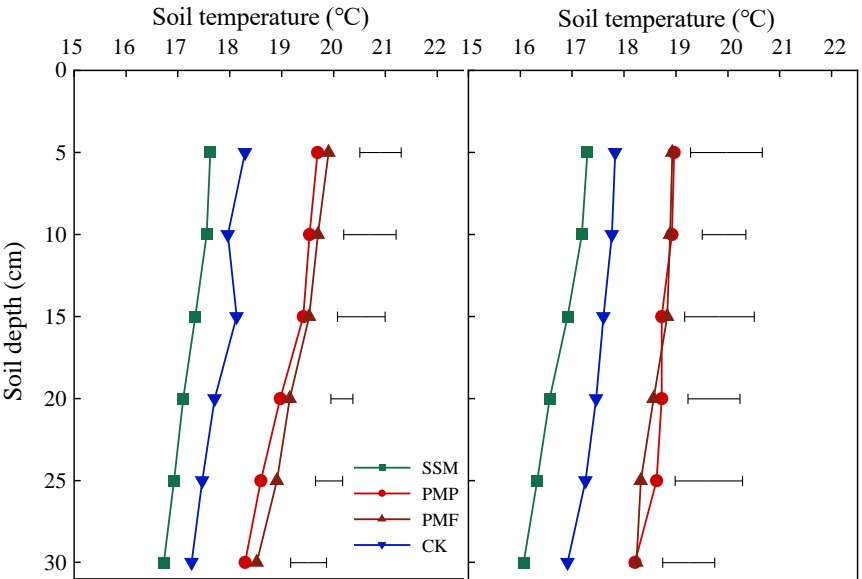

**Figure 5.** Changes in soil temperature in the whole growth period during 2020–2021.

### 3.2.2. Soil Temperature of 0–30 cm Layer at Different Growth Stages

Plastic film mulch significantly increased potato soil temperature from sowing to maturity stage by an average of 1.2 °C, while the average soil temperature of straw strip mulching decreased significantly by 0.6 °C (Figure 6). In the wet year, SSM treatment decreased the soil temperature from sowing to maturity stage by 0.6 °C (0.2–1.1 °C) over CK and plastic film mulching increased potato soil temperature of potato by 1.2 °C (0.3–2.6 °C) compared with CK; the increase showed PMF > PMP, where both PMP and PMF increased soil temperature by 1.2 °C compared to CK, and both increases were greatest at sowing stage. In the normal year, compared with CK, the soil temperature of SSM treatment decreased by 0.6 °C (0.3–1.2 °C) at each growth stage; the decrease was the largest at seedling stage. The soil temperature of plastic film mulching increased 1.1 °C (0.3–2.6 °C) at each growth stage; the increase rate was PMP > PMF. Compared with CK, PMP and PMF increased soil temperature by 1.2 °C and 1.1 °C, and the increase was greatest at the starch accumulation and seedling stage, respectively.

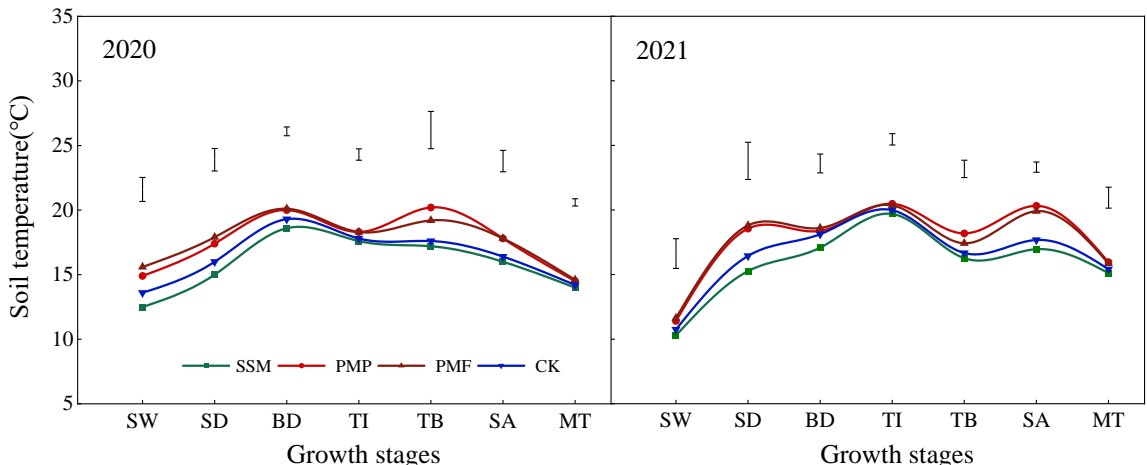

**Figure 6.** Changes in 0–30 cm soil temperature at different growth stages in 2020–2021.

### 3.3. Soil Enzyme Activity

#### 3.3.1. Effects of Different Mulching Treatments on Urease Activity in 0–40 cm Soil Layer

Urease activity varied widely between years under different mulching treatments (Figure 7). Straw strip mulching and plastic film mulching increased by 15.0% and 2.6%, respectively, and the increase is shown as SSM > PMF > PMP. Specifically, in the wet year, compared with CK, the urease activity of SSM treatment was 14.1–16.6% higher than that with CK, while plastic film mulching was not significantly different from CK in each soil layer. In the normal year, SSM increased urease activity by 15.0% and 20.5% in 0–20 and 30–40 cm soil layer over CK, while the difference was not significant in the 20–30 cm soil layer. Urease activity in the 0–10 and 10–20 cm soil layers of PMP and PMF were 10.9% and 9.0%, and 9.5% and 7.4%, respectively, not significantly different from CK in 20-40 cm soil layers. Compared with plastic film mulching, SSM treatment increased urease activity in 0–40 cm soil layer by 11.3% on average.

#### 3.3.2. Effects of Different Mulch Treatments on Catalase Activity in 0–40 cm Soil Layer

The catalase activity of 0–40 cm soil in the two growth seasons varied with different mulching materials (Figure 8). Plastic film mulching increased catalase activity in the whole soil layer by 9.3% on average, with the largest increase in PMF, while straw strip mulching significantly reduced catalase activity by 10.2%. In the wet year, compared with CK, SSM significantly reduced catalase activity in 0–40 cm soil layer by 9.2%, and plastic film mulching increased by 11.1% on average, Catalase activities in 0–40 cm soil layer of PMP and PMF treatments were 6.9–14.6% and 9.9–12.8% higher than that of CK, respectively. In the normal year, the catalase activity in 0–10 and 20–40 cm soil layers of

SSM treatment were 17.1% and 11.8% higher than that of CK. PMP and PMF increased catalase activity in 0–30 cm soil layer by 5.8% and 12.1%, respectively, but there was no significant difference with CK in the 30–40 cm soil layer.

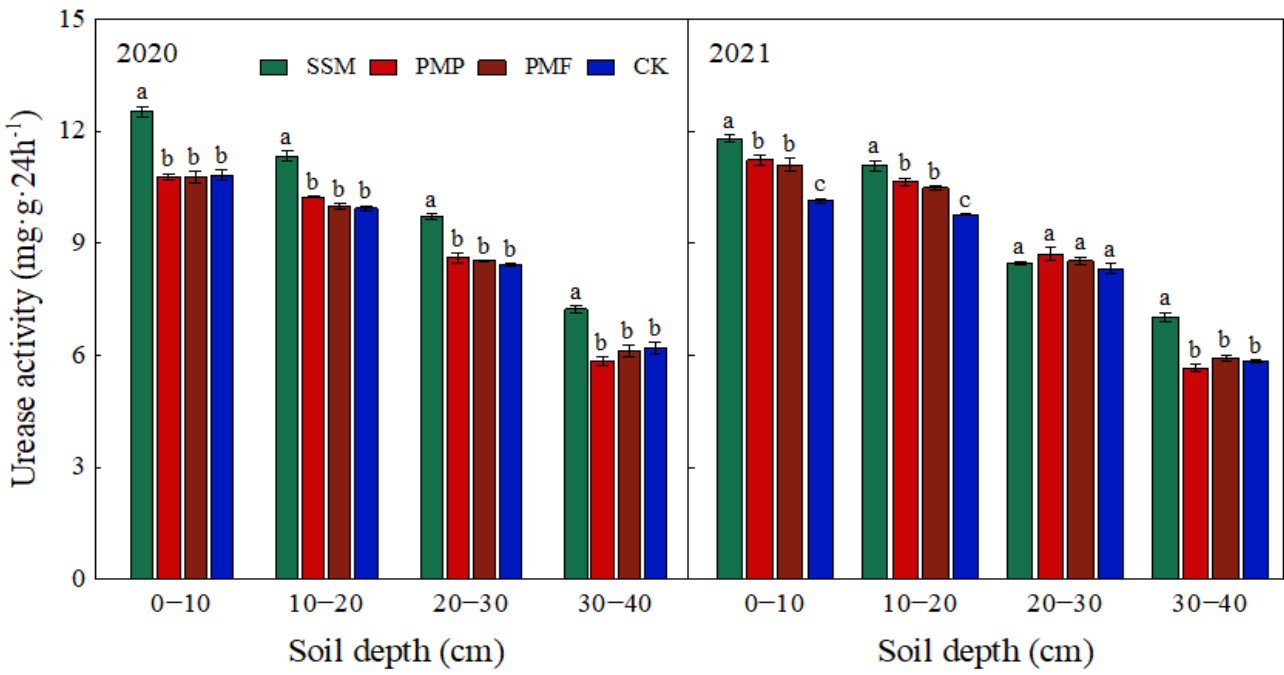

**Figure 7.** Effect of different mulching practices on urease activity of 0–40 cm soil layer. Different lowercase letters indicate significance at 0.05 level ($p < 0.05$).

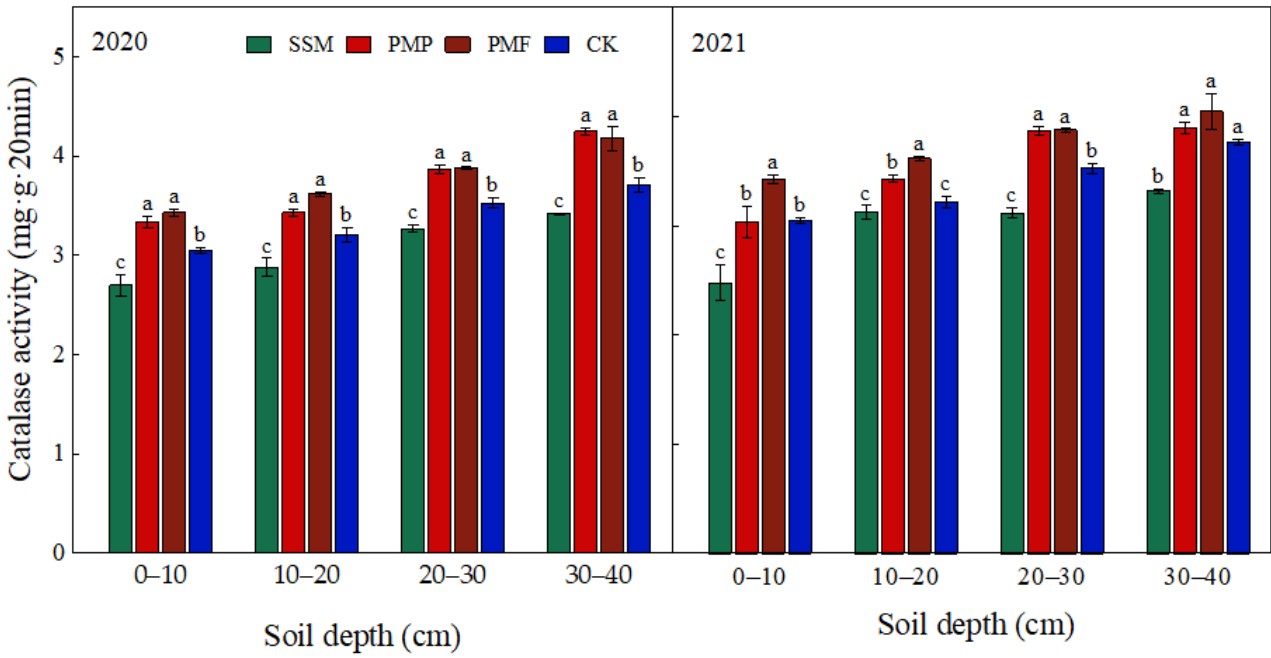

**Figure 8.** Effect of different mulching practices on catalase activity 0–40 cm. Different lowercase letters indicate significance at 0.05 level ($p < 0.05$).

### 3.3.3. Effects of Different Mulch Treatments on Sucrase Activity in 0–40 cm Soil Layer

Mulching practices increased soil sucrase activity in 0–40 cm layer. Straw strip and plastic film mulching increased sucrase activity in the whole soil layer by 19.2% and 7.1%, respectively, and the increase was SSM > PMP > PMF (Figure 9). Specifically, SSM treatment increased soil sucrase activity in 0–40 cm layer by 20.0% compared with CK in the wet year. PMP increased sucrase activity in the 0–30 cm of soil layer by 1.7%, but the difference between PMP and CK treatment were not significant in the 30–40 cm layer. Meantime, sucrase activity in the 0–10 cm and 20–30 cm soil layers with PMF were 15.5% and 12.0% higher than of CK. In normal year, SSM treatment increased sucrase activity by 14.8% in 0–40 cm soil layer, and improved sucrase activity by 9.4% in 0–30 cm soil layer. PMF treatment increased sucrase activity only by 10.7% in 0–10 cm depth and reduced sucrase activity 5.7% in 30–40 cm depth. In the mulching treatments, straw strip mulching was higher by 5.0%, 7.2%, 17.5%, and 21.5% in all soil layers compared to plastic film mulching, respectively.

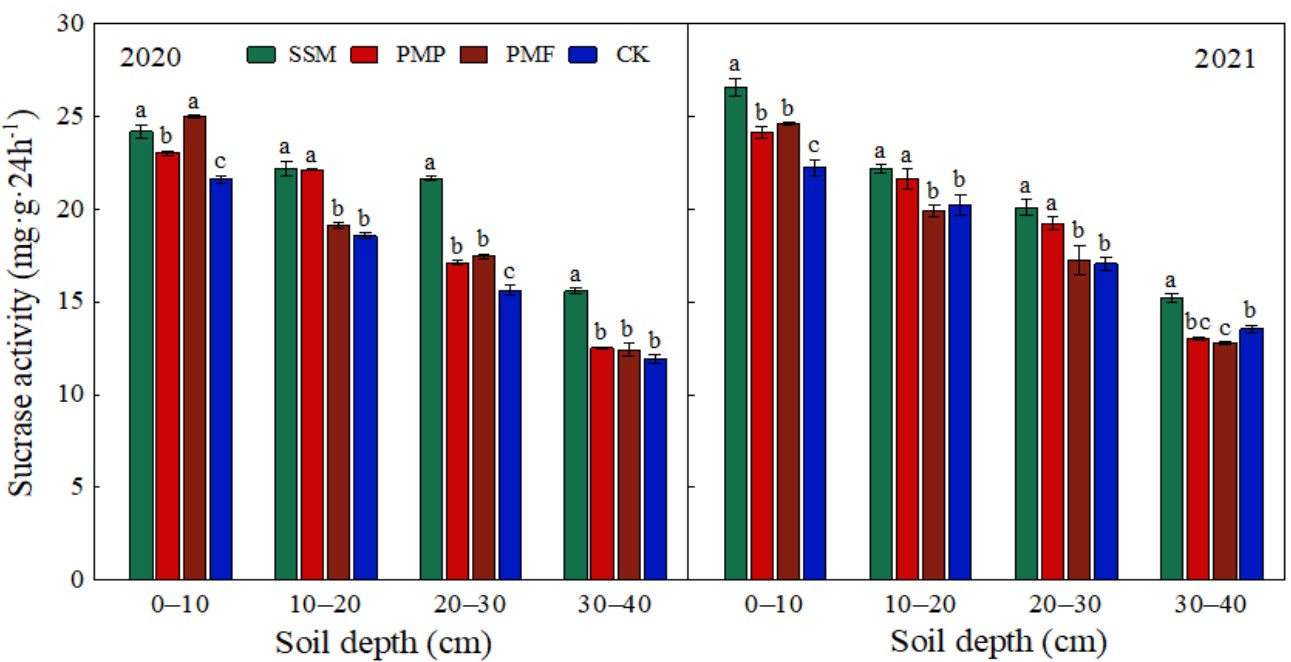

**Figure 9.** Effect of different mulching practices on sucrase activity 0–40 cm. Different lowercase letters indicate significance at 0.05 level (*p* < 0.05).

### 3.4. Changes in Yield under Different Mulch Treatments

The treatments × year interaction had non-significant effect of on potato weight yield, while treatments × year interaction revealed the significant effect on water use efficiency. The yield of fresh tuber and water use efficiency were significantly increased by mulching treatments in two growing seasons compared with the control; the increase was plastic film mulching > straw strip mulching (Table 1). In the wet year, plastic film mulching and straw strip mulching increased fresh tuber yield by 26.5% and 13.7%, respectively, of which PMP and PMF increased by 26.4% and 26.6%, respectively, compared with CK, but there was no significant difference between PMP and PMF. SSM, PMP, and PMF significantly increased water use efficiency by 100.3%, 94.4%, and 113.9%, respectively. In the normal year, plastic film mulching and straw strip mulching increased fresh potato tuber yield by an average of 36.2%, and 24.8% and enhanced water use efficiency by an average of 26.5% and 27.2%, respectively. Of particular note, PMP and PMF increased fresh potato tuber yield by 38.9% and 33.5% and enhanced water use efficiency by 27.4% and 26.1%, respectively, compared with CK. There was a significant difference between PMP and PMF in improving fresh tuber yield, but no significant difference in water use efficiency.

**Table 1.** Tubre yield and its components for different mulching treatments during 2020–2021.

| Year | Treatment | Potato Weight Yield (kg ha$^{-1}$) | WUE (kg ha$^{-1}$ mm$^{-1}$) | Tuber Number Per Plant | Weight per Fresh Tuber (g) | Commodity Rate (%) |
|---|---|---|---|---|---|---|
| 2020 | SSM | 47,217.9 b | 31.1 a | 9.8 a | 92.1 b | 82.5 a |
| | PMP | 52,487.4 a | 30.2 a | 10.4 a | 98.8 a | 84.9 a |
| | PMF | 52,581.1 a | 33.3 a | 10.3 a | 99.1 a | 85.3 a |
| | CK | 41,532.3 c | 15.6 b | 8.9 b | 88.4 c | 73.5 b |
| 2021 | SSM | 40,963.2 c | 26.5 b | 7.6 ab | 121.0 a | 90.2 a |
| | PMP | 45,583.1 a | 27.4 a | 8.1 a | 107.9 b | 87.7 b |
| | PMF | 43,821.6 b | 27.0 ab | 8.3 a | 103.6 bc | 84.5 c |
| | CK | 32,817.3 d | 22.8 c | 6.9 b | 95.3 c | 85.0 c |
| T | | ** | ** | ** | ** | ** |
| Y | | ** | NS | ** | ** | ** |
| T × Y | | NS | ** | NS | ** | ** |

NS shows no significance at 0.05 level. ** shows significance at 0.01 level. Different lowercase letters indicate significance at 0.05 level ($p < 0.05$).

The treatments × year interaction was not significant on tuber per plant, while treatments × year interaction revealed the significant effect on weight per fresh tuber and commodity. In terms of yield components, on average over two growth seasons, mulching cultivation practices significantly improved tuber number per plant and weight per tube by 10.1–20.3% and 4.1–27.0%, respectively, compared with CK; there was no significant difference in tuber number per plant among mulching treatments, while the weight per tuber under each mulch cultivation varied between years. PMP and SSM treatments were the highest and increased weight per tuber by 12.1% and 27.0%, respectively, in the normal year. The increase in weight per tuber further promoted the increase in commodity rate. In the wet year, the commodity rate of SSM, PMP, and PMF increased by 12.2%, 15.5%, and 16.0%, respectively, but the differences between mulching cultivation practices were not significant. The commodity tube rates of SSM and PMP were 6.1% and 3.1% higher than CK in normal year, respectively.

*3.5. Correlation between Tuber Yield and Soil Moisture, Temperature and Enzyme Activity under Different Mulching Treatments*

Tuber number per plant (TN) emerged as the key factor influencing variations in tuber yield across various planting layouts and environmental conditions (Figure 10). The correlation analysis showed that the fresh tuber yield (TY) was significantly positively correlated with STW (r = 0.95, $p < 0.01$), SWS (r = 0.92, $p < 0.01$), WUE (r = 0.85, $p < 0.01$), and ST (r = 0.65, $p < 0.05$). According to the components of tuber yield, the tuber number per plant was significant positively correlated with SWS (r = 0.89, $p < 0.01$), WUE (r = 0.84, $p < 0.01$), and ST (r = 0.64, $p < 0.05$). There was a very significant positive correlation between weight per tuber and SWS (r = 0.71, $p < 0.01$), URE (r = 0.72, $p < 0.01$), and SAC (r = 0.94, $p < 0.01$). Soil moisture and temperature condition could affect soil enzyme activity, in which soil moisture was significantly positively associated with SAC (r = 0.66, $p < 0.05$), and soil temperature was significantly positively associated with SCAT (r = 0.94, $p < 0.01$). Findings showed that mulching coordinated the soil moisture and temperature conditions, improved the soil enzyme activity, and, finally, promoted the tuber yield.

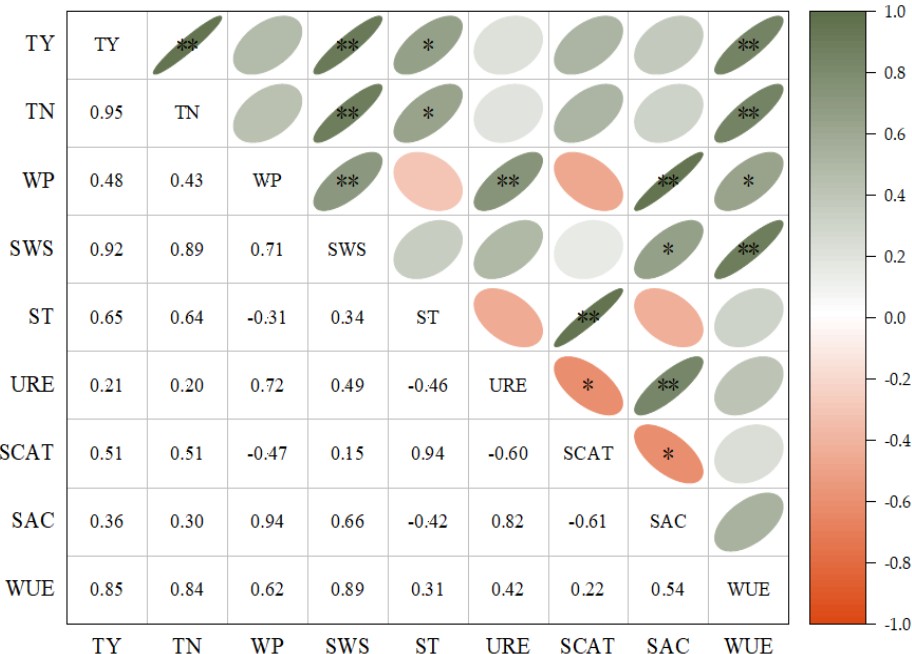

**Figure 10.** Correlation of tuber yield with soil hydrothermal and enzyme activities under different mulching treatments. TY: tuber yield; TN: tuber number per plant; WP: weight per fresh tuber; SWS: average soil moisture throughout the growing season; ST: soil temperature at 0–30 cm during the whole growing stage; URE, SCAT, and SAC: average soil urease, catalase, and sucrase activity of 0–40 cm soil at seeding, tuber bulking, and maturity stage; WUE: water use efficiency. Statistical significance denoted by * $p \leq 0.05$; ** $p \leq 0.01$.

## 4. Discussion

### 4.1. Effects of Mulching on Soil Moisture and Temperature

Surface mulching can effectively reduce evaporation of soil moisture and increase the capacity of soil moisture retention [21], thereby addressing the conflict between moisture requirements of crops and the availability of moisture in the soil [22,23]. Liang et al. [24] found that plastic film mulching could effectively increase the soil moisture of potato plough layer by 27.4%. Pu et al. [25] reported that moisture leakage and non-physiological crop evaporation contributed to the relatively large changes in soil moisture in the upper and middle layers under the mulching practices due to root system uptake of moisture from deeper layers. We found that mulching treatments were effective in harvesting rainfall and supplementing moisture in different soil layers that could provide moisture retention, and the greatest increase was in the middle soil layer. In addition, this study found that there were differences in soil moisture in different growth stages of potato under mulching cultivation practices between two experimental years. The soil moisture of PMP was highest throughout the growing season in normal year. During the wet year, the SSM treatment showed lower soil moisture compared to plastic film mulching from sowing to tuber expansion. However, as the potato growth progressed, the soil moisture of SSM treatment surpassed that of plastic film mulching, which, due to the fluctuations in soil moisture throughout the growing season, were intricate, influenced by factors such as interannual precipitation, soil evaporation, mulching material, and mulching intensity [7]. The effective precipitation at this stage from starch accumulation to maturity was 87.9 mm in the wet year and the effective precipitation at this stage was 12.2 mm in the normal year. Although plastic film mulching could effectively prevent the evaporation of surface moisture, at the same time, the higher coverage hinders the infiltration of some rainwater [26]. We are concerned that using plastic film mulching for an extended period can impact the soil's moisture balance by decreasing the infiltration of rainwater. On the one hand, the semi-enclosed straw strip mulching could accumulate precipitation in time; on the other

hand, its flocculent and thicker waxy surface make it have better moisture absorption and moisture retention performance [27]. Therefore, under the condition of abundant precipitation, straw strips mulching has a superior impact on retaining moisture during the advanced growth phase of potatoes.

Appropriate soil temperature is the basis for ensuring crop growth; it is also the key to promoting plant root activity [28]. Mulching cultivation practice not only increases soil moisture, but also adjusts soil temperature in the plowed layer [29]. Chang et al. [30] noted that straw mulching decreased in soil temperature during the middle stage of potato development (tuber expansion stage) to 18.3–21.6 °C, which was more beneficial for potato to growth. Lu et al. [31] indicated that plastic mulching led to an increase in average soil temperature at depths of 5–15 cm by 1.5 °C and 1.2 °C compared to straw mulching and no mulching throughout the entire growth season, respectively. In this study, plastic film mulching significantly increased soil temperature at all growth stages of potato, while straw strip mulching significantly decreased soil temperature, which was consistent with earlier testing. Soil temperature distribution depends on the radiation absorption, reflection, and permeation [32]. Plastic mulching can make the light reach the ground directly, thus increasing the magnitude of daytime warming, reducing energy fluxes, and allowing the soil to absorb and store more temperature [33]. However, straw mulching has higher reflectivity and lower thermal conductivity, it blocks solar radiation, and it makes the surface receive less temperature, so the cooling effect is significant [7]. In the present study, the cooling effect of straw strips mulching occurred during the reproductive growth stage of potato, which was reduced by 0.2–1.1 °C compared to CK; growth and yield formation were not affected by the slightly lower soil temperature at this stage [34].

### 4.2. Effect of Mulch on Soil Enzyme Activity

As a product of soil microbial activities, soil enzymes participate in soil biochemical processes. Their activity is influenced by soil hydrothermal and aeration conditions, which make it sensitive to environmental changes [35]. The results of the study indicated that the soil urease and sucrase activities showed "surface aggregation"; soil urease and sucrase activities experienced a gradual decrease as the soil layer deepened throughout the entire potato growth period, which is comparable to the results of Hechim et al. [36]. On the one hand, the surface soil was in direct contact with solar radiation, so the soil temperature was higher, and the microbial activity was more frequent [37]; on the other hand, the surface soil was loose, and potato had more root exudates, which increases soil fertility and promotes root growth [38]. Therefore, urease and sucrase activities were increased in the surface layer. For catalase, deeper soils are compact and less permeable, so the redox environment contains a more abundant substrate, hydrogen peroxide, resulting in higher catalase activity of deeper soil than shallower soil [39]. Yao et al. [40] found that furrow and ridge plastic film mulching could increase the activities of urease and catalase in potato field, reduce soil pH, improve the physical and chemical properties of rhizosphere soil, and increase the potato yield. Akhtar et al. [41] discovered that straw mulching led to a significant rise in the levels of invertase, urease, alkaline phosphatase, and catalase in soil ranging from 0 to 40 cm deep. Studies also showed that compared with straw mulch, plastic mulch is more conducive to increasing the abundance of soil proteobacteria in the early growth period of potato to improve urease activity and sucrase activity, thus catalyzing the metabolism of organic matter in the soil and promoting the decomposition and transformation of organic nitrogen and organic matter [19]. In this experiment, mulch cultivation practices increased the activities of soil urease and sucrase in the whole depth, and the effect of straw strip mulching was the best; this could be due to the fact that straw mulching increased the microbial population and microbial mass C or N, which provided organic matter as a substrate for soil enzymes, contributing to a positive enzymatic reaction, thus increasing soil enzyme activity [14]. The extended use of straw mulching and its subsequent breakdown on the soil surface has shown positive effects on fertilization and moisture retention. This practice provides ample nutrients and creates a

favorable hydrothermal environment for soil microorganisms to thrive, thus enhancing the microbial population and increasing soil enzyme activity [42]. The physical and chemical properties of soil were altered by fluctuations in soil temperature, potentially resulting in a lack of nutrients during the later stages of potato growth due to elevated temperatures under plastic mulch [33]. In addition, straw strip mulching decreased catalase activity in two growing seasons. This is because catalase was sensitive to soil temperature, and soil warming could increase the rates of microbial reproduction and metabolism, produce more enzymes to participate in the carbon and nitrogen cycle, expand the enzyme pool, and then increase the enzyme activity [43]. However, the cooling effect of straw strip mulching inhibits soil respiration, resulting in the passivation of catalase activity [44]. In practice, environmental factors and management strategies can have an impact on soil microbial metabolism and enzyme activity.

*4.3. Effect of Mulching on Tuber Yield*

Mulching can effectively regulate soil microenvironment conditions during the crop's reproductive period, which in turn affects soil enzyme activity and ultimately guarantees high tuber yield [45]. Chen et al. [46] discovered that the mulched treatments increased tuber yield by 36.9–61.2% and water use efficiency by 38.7–45.5% over traditional no mulching. Ma et al. [47] pointed out that mulching improves moisture capacity, raises soil temperature, and promotes an increase in soil alkaline phosphatase activity and sucrase activity, ultimately increasing yields by an average of 32.5%. We found that three mulching practices significantly increased fresh tuber yields by 13.7–38.9% in a two-year trial, and that plastic mulching was greater than straw strip mulching. This is mainly due to the fact that mulching provided better moisture and temperature for crop growth in dry conditions, but soil enzyme activity was higher under straw mulch, which encourages the decomposition and cycling of soil nutrients and provides rich nutrients for crop growth [16]. The tuber yield in the wet year was greater than in the normal year, which was mainly attributed to the higher precipitation in the wet year [48]. The increase in tuber number per plant mainly accounted for the improved yield due to mulching compared to CK under the identical planting density. Therefore, mulching facilitates tuber formation by providing a stable moisture supply, which further enhances the tuber's use of soil moisture and has a significant increase in tuber yield.

Hou et al. [49] demonstrated that the effect of soil moisture on wheat yield showed a direct contribution. Wu et al. [50], through the analysis of potato yield and water–temperature–fertilizer flux pathway, found that soil moisture under different treatments had the greatest effect on potato yield. The study revealed that the soil moisture under various mulch treatments influenced the yield of potatoes. Temperature was identified as the next most influential factor, with the different mulching techniques enhancing soil enzyme activities and improving the hydro-thermal conditions of the soil. These enhancements ultimately resulted in a notable increase in potato yield.

## 5. Conclusions

Mulching practices could increase the soil moisture at 0–200 cm by 8.2% during potato growth, compared to the no mulching with conventional planting; the increase rate in the wet year was higher than in the normal year, and plastic film mulching was greater than straw strip mulching. Plastic film mulching had a significant warming effect compared to no mulching, while straw strip mulching had a significant cooling effect. The activities of urease and sucrase in soil under mulching practices were higher than no mulching, and were mostly processed by straw strip mulching; plastic mulching significantly increased soil catalase activity, and straw strip mulching significantly decreased catalase activity. Mulching increased the tuber number per plant, which in turn significantly increased the fresh tuber yield by 30.8% for straw strips mulching and 18.6% for plastic mulching over CK. In the rainfed agricultural area of Northwest China, straw strip mulching, as a

sustainable and protective mulching measure, is conducive to improving the ecology of agricultural land and the development of green agriculture.

**Author Contributions:** Data curation, formal analysis and writing–original draft, W.S.; writing—review and editing, L.C.; formal analysis, F.H. and H.C.; project administration, Z.B.; methodology, Y.C.; data analysis, L.W.; investigation, C.H. All authors have read and agreed to the published version of the manuscript.

**Funding:** This study was sponsored by the National Key Research and Development Program (2021YFD1900700; 2022YFD2001304), the Industrial Support Program of Gansu Provincial Colleges and Universities (2022CYZC-48), the Science and Technology Plan Project of Gansu Province (22CX8NA046), and the National Natural Science Foundation of China (31960239).

**Data Availability Statement:** The data presented in this study are available on request from the corresponding author.

**Conflicts of Interest:** The authors declare no conflict of interest.

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
