# Peer review of "Mulching Practices Improve Soil Moisture and Enzyme Activity in Drylands, Increasing Potato Yield"

_agronomy, doi:10.3390/agronomy14051077_

Round 1
Reviewer 1 Report
Comments and Suggestions for Authors
see in attached file

Comments on the Quality of English LanguageAuthor Response
Dear Reviewer:
Thanks very much for taking your time to review this manuscript. I really appreciate all your comments and I have benefited a lot from the comments on the manuscript. After receiving your comments on returning the manuscript for revision, we have carefully analyzed and revised the manuscript, criticize and correct any shortcomings or mistakes that may occur. The following are the detailed corrections to our manuscript entitled “Mulching practices improve soil moisture and enzyme activity in drylands, increasing potato yield” (ID: agronomy - 2998306).
Abstract
Comments 1: Add clear hypothesis and objective of the study. Also conclude the study in comprehensive way. Comments 1: Thank you for pointing this out. We agree with this comment. Therefore, we have added objective of the study that [However, the effects of different mulching materials on soil hydrothermal, enzyme activity and yield in potato fields have not been systematically understood. Thus,] and clear hypothesis [Study have found that the enhancement of soil hydrothermal conditions resulting from the use of plastic film mulch and straw strip mulch plays a vital role in increasing potato yields. Does the soil hydrothermal environment affect soil enzyme activity? How do mulching practices impact soil hydrothermal conditions, enzyme activity and tuber yields?]. In the revised manuscript this change can be found –page 1, line 17 - 19. page 3, line 102 - 106.
Introduction
|
Comments 2: The study acknowledges the positive impacts of mulching on crop yields and water productivity, it lacks specific data or references to support these claims |
Response 2: Thank you for pointing this out. We have modified. In our introductory parts, we specifically write in lines 59-60 that “[the average yield of wheat, maize and potato increased by 24.3%, and the water use efficiency increased by 27.6%]” and we have references. In the revised manuscript this change can be found –page 2, line 59 - 60. |
Comments 3: It briefly mentions the negative consequences of plastic film accumulation in the soil without delving into the extent of this issue or discussing potential mitigation strategies. |
Response 3: We sincerely appreciate the valuable comments. We agree with this comment. In the introduction, we described the negative effects of mulching on potato growth restrains and restrain the sustainable development of agriculture ultimately. We added the benefits of straw mulch treatments. [Straw mulching is a straightforward and user-friendly method, which is more consistent with green agriculture]. In the revised manuscript this change can be found – page 2, line 62 - 64. Comments 4: You mentioned that how straw strip mulching is seen as a good option, but it doesn't give proof Response 4: Thank you for pointing this out. We agree with this comment. Therefore, we added the introduction to straw strip mulching. [Straw strip mulching technology is to divide the plot into planting strip and mulching strip, the two strips are arranged alternately, the whole corn stalks are covered in the mulched strip, and potatoes are sown in the planting strip.]. In the revised manuscript this change can be found – page 2, line 67-69. Comments 5: Given the importance of soil enzymes in soil health and their relationship with soil hydrothermal conditions. It outlines the roles of specific enzymes like urease, catalase, and sucrase in soil nutrient cycling and plant growth. However, it lacks a clear demonstration of how mulching practices affect soil enzyme activity and subsequent potato yield. Response 5: Thank you for your suggestion. We agree with this comment. Therefore, we have explained demonstration in discussion section. [That furrow and ridge plastic film mulching could increase the activities of urease and catalase in potato field, reduce soil pH, improve the physical and chemical properties of rhizosphere soil and increase the potato yield] and [Mulching can effectively regulate soil microenvironment conditions during the crop's reproductive period, which in turn affects soil enzyme activity and ultimately guarantees high tuber yield]. In the manuscript page 14, line 441-443 and 471-473. Comments 6: Previous studies indicating the positive influence of moisture and temperature on enzyme activity, it doesn't provide specific evidence linking these factors to the chosen mulching techniques or their impact on potato yield. Response 6: Thank you for pointing this out. We agree with this comment. Therefore, we have added specific evidence of previous studies and explained them further in the discussion. [Yang, et al. pointed out at the tuber swelling stage, sucrase and catalase activities of plastic film mulch were significantly higher than of straw mulch, which sucrase activities increased by 18.71% and catalase activities increased by 17.44%, the rate of large potatoes and yield were increased by 20. 15% and 17. 90% more than CK] and [Studies also showed that compared with straw mulch, plastic mulch is more conducive to increasing the abundance of soil proteobacteria in the early growth period of potato to improve urease activity and sucrase activity, thus catalyzing the metabolism of organic matter in the soil and promoting the decomposition and transformation of organic nitrogen and organic matter].In the revised manuscript this change can be found –page2, line 92 - 95 and page 13, line 446 - 450. Comments 7: Importance of soil enzymes and mulching in agriculture is mentioned but it lacks specificity and clarity regarding the research objectives and methodology. Response 7: Thank you for pointing this out. We agree with this comment. Therefore, we have modified. We added the purpose of the study. [Therefore, the purpose of this study is (1) to analyze and compare the effects of different mulch treatments on potato soil hydrothermal conditions, soil enzyme activities and potato yield; (2) clarify the mechanism of soil hydrothermal effects on enzyme activity differences under different mulch treatments; (3) assessment of the application value of various mulching techniques for potato cultivation in the semiarid rainfed farming system] and deleted sentence that “Our research provides a feasible theoretical basis for exploring the best way of green mulch suitable for local production and ecology”. Mention exactly where in the revised manuscript this change can be found – page 3, line 107-113. Comments 8: Novelty in the study should be clearly mention. Response 8: We sincerely appreciate the valuable comments. We agree with this comment, we have changed “Therefore, a two-year field experiment was carried out in the rain-fed agricultural area of Northwest China.” to [“To explored the effects of mulching practices on soil hydrothermal, enzyme activities and tuber yield, to improve the relationship between hydrothermal and nutrient requirements for potato growth and agronomic practices”] In the revised manuscript this change can be found – page 2 - 3, line 98 - 102.
Materials and Methods
Comments 9: It mentions the classification of the experimental years based on precipitation and drying index, it does not clearly explain how these classifications influenced the research findings or the interpretation of results. Response 9: Thank you for your suggestion. We agree with this comment. DI is the division standard, DI=(P-M) / σ, where DI is the drought index (DI < -0.35 is a drought year; DI > 0.35 is an abundant water year; -0.35 ≤ DI ≤ 0.35 is a shallow water year); P is the current year's rainfall (mm); M is the multi-year average rainfall (mm); and σ is the multi-year average rainfall mean squared error. In this study, the total amount of rainfall during the reproductive period of potato was divided into: rainfall in 2020 was 352.6 mm, with a DI of 0.63, which belonged to a wet year; rainfall in 2021 was 284.5 mm, with a DI of - 0.27, which belonged to a normal year. Dryness in land surfaces simply refers to the lack of soil moisture. Differences in the effect of different mulch materials on soil moisture in the later potato growing season are due to differences in precipitation year patterns, which we explain in the Discussion section. We explained that the difference in potato fresh potato yield between the two years was mainly due to soil moisture. Comments 10: It is 2 years study. In the statistical analysis, it should be clearly mentioned that why you not pool the data and what about the year effect. Because you mentioned a significant difference in the rainfall in these two years. Response 10: Thank you for your suggestion. We agree with this comment, we have added an interaction between treatment and year in Table 1, further added explanatory descriptions and note [The treatments × year interaction had non-significant effect of on potato weight yield, while treatments × year interaction revealed the significant effect on water use efficiency] and [The treatments × year interaction was not significant on tuber per plant, while treatments × year interaction revealed the significant effect on weight per fresh tuber and commodity]. The note that [NS shows no significance at 0.05 level. ** shows significance at the 0.01 level.] In the revised manuscript this change can be found – page10-11, line 328 - 329, 342 – 343, and line 354 (Table 1).
Results and discussion
Comments 11: Results are clearly presented and written however discussion need much improvement. In discussion you only presented the results of previous study. Authors should improve the discussion with clear reasoning of your results. For example you do not address potential interactions between mulching treatments and other agronomic practices, such as irrigation management or fertilizer application, which could influence the outcomes observed. Response 11: Thank you for pointing this out, we agree with this comment, we have provided additional clarification in the discussion section. [The extended use of straw mulching and its subsequent breakdown on the soil surface has shown positive effects on fertilization and moisture retention. This practice provides ample nutrients and creates a favorable hydrothermal environment for soil microorganisms to thrive, thus enhancing the microbial population and increasing soil enzyme activity] and [In practice, environmental factors and management strategies can have an impact on soil microbial metabolism and enzyme activity] In the revised manuscript this change can be found – page14, line 455 - 459, 467 – 469. We appreciate your suggestion, language problems in the manuscript have been carefully corrected.
we would like to express our great appreciation to you and reviewers for comments on our paper.
|
Reviewer 2 Report
Comments and Suggestions for Authors
Thank you for this research titled "Mulching practices improve soil moisture and enzyme activity in drylands, increasing potato yield". We come across some research on plastic mulch in some colors and different plant mulches. Frankly, it is not a new research for the world in general. However, I find the research important because it is a region of China and is important for that region. In addition, when considered together with climate changes, it is also important to combine different mulching practices for direct crop-based cultivation. I think the research was conducted well. However, I have some suggestions and corrections. I noted these on the manuscript. It will be better if it is taken into consideration.
Line 55-56: This sentence seems unrelated to the fluency of the sentences above. It is written as if it were the result of a newly completed research, followed by general statements. It should be checked.
Line 59-61: The sentence flow should be checked, a different topic is tried to be conveyed after the previous sentence.
Line 82: What is meant by this expression? uptake of plant nutrients?
Line 94: Although the research on this subject is stated to be insufficient, there are other studies around the world on different mulches. Therefore, sample research citations can be made from some mulching studies just before the purpose of the research.
Line 115: Which potato variety was used in the research?
Line 116-118: I think that the use of "it set...." in this sentence, which talks about the applications in the essay, is not appropriate. A meaning that does not fully encompass has been formed.
Line 119: There is no predicate in this sentence.
Line 119: T
Line 122: R
Line 125: L
Line 127: It doesn't sound good semantically to start like this.
Line 127: F
Line 145: s or .
Line 213: One has an abbreviation and the other does not. Readers may be confused.
Line 224: This statement is not appropriate. The article is evaluated here.
Line 226: rainy
Line 262: soil layer
Line 294: soil layer
Line 301: Is this word necessary? here and in other places....
Line 314: treatments
Line 315: rainy
Line 321: This usage is in the passive form. it should be checked.
Line 321: A different word can be used.
Line 342: (TN) was
Line 378: found or was found?
Line 436: Here and elsewhere, attention should be paid to the use of capital letters after the comma.
Line 578: italic

Comments on the Quality of English LanguageThank you for this research titled "Mulching practices improve soil moisture and enzyme activity in drylands, increasing potato yield". We come across some research on plastic mulch in some colors and different plant mulches. Frankly, it is not a new research for the world in general. However, I find the research important because it is a region of China and is important for that region. In addition, when considered together with climate changes, it is also important to combine different mulching practices for direct crop-based cultivation. I think the research was conducted well. However, I have some suggestions and corrections. I noted these on the manuscript. It will be better if it is taken into consideration.
Line 55-56: This sentence seems unrelated to the fluency of the sentences above. It is written as if it were the result of a newly completed research, followed by general statements. It should be checked.
Line 59-61: The sentence flow should be checked, a different topic is tried to be conveyed after the previous sentence.
Line 82: What is meant by this expression? uptake of plant nutrients?
Line 94: Although the research on this subject is stated to be insufficient, there are other studies around the world on different mulches. Therefore, sample research citations can be made from some mulching studies just before the purpose of the research.
Line 115: Which potato variety was used in the research?
Line 116-118: I think that the use of "it set...." in this sentence, which talks about the applications in the essay, is not appropriate. A meaning that does not fully encompass has been formed.
Line 119: There is no predicate in this sentence.
Line 119: T
Line 122: R
Line 125: L
Line 127: It doesn't sound good semantically to start like this.
Line 127: F
Line 145: s or .
Line 213: One has an abbreviation and the other does not. Readers may be confused.
Line 224: This statement is not appropriate. The article is evaluated here.
Line 226: rainy
Line 262: soil layer
Line 294: soil layer
Line 301: Is this word necessary? here and in other places....
Line 314: treatments
Line 315: rainy
Line 321: This usage is in the passive form. it should be checked.
Line 321: A different word can be used.
Line 342: (TN) was
Line 378: found or was found?
Line 436: Here and elsewhere, attention should be paid to the use of capital letters after the comma.
Line 578: italic
Author Response
Dear reviewer:
Thanks very much for taking your time to review this manuscript and you scrutinize. I really appreciate all your comments and I have benefited a lot from the comments on the manuscript. After receiving your comments on returning the manuscript for revision, we have carefully analyzed and revised the manuscript, criticize and correct any shortcomings or mistakes that may occur. The following are the detailed corrections to our manuscript entitled “Mulching practices improve soil moisture and enzyme activity in drylands, increasing potato yield” (ID: agronomy - 2998306).
Comments 1: This sentence seems unrelated to the fluency of the sentences above. It is written as if it were the result of a newly completed research, followed by general statements. It should be checked. |
Response 1: Thank you for pointing this out. We agree with this comment. Therefore, we have changed [As a result, the average yield of wheat, maize and potato increased by 24.3%, and the water use efficiency increased by 27.6%] to [and the average yield of wheat, maize and potato increased by 24.3%, the water use efficiency increased by 27.6%]. In the revised manuscript this change can be found – page 2, and line59 - 60. |
Comments 2: The sentence flow should be check, a different topic is tried to be conveyed after the previous sentence. |
Response 2: Thank you for pointing this out. We agree with this comment. Therefore, we have modified. We added [“Straw mulching is a simple and easy to operate technique, which is more consistent with green agriculture, although”]. In the revised manuscript this change can be found – page2, line 62 - 64. |
Comments 3: What is meant by this expression? uptake of plant nutrients? |
Response 3: We sincerely appreciate the valuable comments. Mulching improves the hydrothermal environment of the soil, which in turn improves the nutrient status of the soil and promotes plant nutrient uptake. |
Comments 4: Although the research on this subject is stated to be insufficient, there are other studies around the world on different mulches. Therefore, sample research citations can be made from some mulching studies just before the purpose of the research. |
Response 4: Thank you for pointing this out. We agree with this comment. Therefore, we have modified. [The purpose of this study is (1) to analyze and compare the effects of different mulch treatments on potato soil hydrothermal conditions and, soil enzyme activities and potato yield; (2) clarify the mechanism of soil hydrothermal effects on enzyme activity differences under different mulch treatments; (3) assessment of the application value of various mulching techniques for potato cultivation in the semiarid rainfed farming system.] and deleted sentence that “Our research provides a feasible theoretical basis for exploring the best way of green mulch suitable for local production and ecology”. Mention exactly where in the revised manuscript this change can be found – page 3, line 107-113. |
Comments 5: Which potato variety was used in the research? |
Response 5: Thank you for the suggestion. We have added the information about potato variety. [The tested material is “Longshu 7”, which belong to medium-late maturing variety]. In the revised manuscript this change can be found – Page4, line 136. |
Comments 6: I think that the use of "it set...." in this sentence, which talks about the applications in the essay, is not appropriate. A meaning that does not fully encompass has been formed. |
Response 6: We apologize for our carelessness. Based on your reminder, “It set” changed “Four treatments were applied in all experiments”. In the revised manuscript this change can be found – page 4, Line 137-138. |
Comments 7: There is no predicate in this sentence. |
Response 7: Thank you for pointing this out. We agree with this comment. Therefore, we have change “the plots with area of 80 m2 (16 m × 5 m) with three replications” to [each treatment was replicated three times in a randomized block design, with each plot area 80 m2 (16 m×5 m)]. In the revised manuscript this change can be found – page4, line 141-142. |
Comments 8: Check writing” t, r, l, f”. |
Response 8: We were really apologized for our careless mistakes. Thank you for your reminder, we changed “t” to “T”, “r” to “R”, “l” to “L” and “f” to “F”. In the revised manuscript this change can be found – page 3, line 142, 145, 148, 150. |
Comments 9: It doesn't sound good semantically to start like this. |
Response 9: We sincerely thank the reviewer for careful reading. As suggested by reviewer, “Sow 2 rows per ridge” changed [“Two rows of per ridge”]. In the revised manuscript this change can be found – page4, line150. |
Comments 10: One has an abbreviation and the other does not. Readers may be confused. |
Response 10: According to the comments of the reviewers. Changed “SSM” to [“Straw strip mulching”]. In the revised manuscript this change can be found – page 6. line 243. |
Comments 11: This statement is not appropriate. The article is evaluated here. |
Response 11: Thank you for pointing this out. We agree with this comment. Therefore, we have deleted “Generally speaking”. In the revised manuscript this change can be found – page 7, line 253. |
Comments 12: rainy |
Response 12: The year of experimentation was classified according to the precipitation the whole growth period of potato and drying index. The year of 2020 were identified as wet year. |
Comments 13: Add “layer” in the title |
Response 13: We have explained the change made, including the exact location where the change can be found in the revised manuscript. In the revised manuscript this change can be found – page 8 ,9 and 10, line 284, 299 and 313. |
Comments 14: Is this word necessary? here and in other places.... |
Response 14: We sincerely thank the reviewer for careful reading. As suggested by reviewer, we have deleted “that”. In the revised manuscript this change can be found – page10, line 331. |
Comments 15: Modify the word treatment |
Response 15: We sincerely thank the reviewer for careful reading. As suggested by reviewer, we have changed “treatment” to “treatments”. Thank you for your reminder. In the revised manuscript this change can be found – page 11, line 320. |
Comments 16: This usage is in the passive form. it should be checked. |
Response 16: We have corrected the grammatical problems based on the reviewer's comments. Deleted “were” in sentence. In the revised manuscript this change can be found – page 11, line 338. |
Comments 17: A different word can be used. |
Response 17: According to the comments of the reviewers, we have changed “improve” to “enhance”. In the revised manuscript this change can be found – page 10, line 338. |
Comments 18: (TN) was |
Response 18: Thank you for pointing this out. We agree with this comment. In this article, TN point to tuber number per plant. We have explained in the diagram. In the revised manuscript this change can be found – page 12, line 372 (Figure 10). |
Comments 19: found or was found? |
Response 19: We were really apologized for our careless mistakes. Thank you for your reminder, we changed “found” to [“was found”]. In the revised manuscript this change can be found – page 12, line 389. |
Comments 20: Here and elsewhere, attention should be paid to the use of capital letters after the comma. |
Response 20: We apologize for our carelessness. Based on your reminder, we changed “This” to [“this”]. In the revised manuscript this change can be found – page 14, line 477. Comments 21: italic Response 21: Thank you for pointing this out, we apologize for our carelessness. We have modified it in the manuscript. In the revised manuscript this change can be found – page 16, line 608. |
|
We would like to express our great appreciation to you and reviewers for comments on our paper.
Reviewer 3 Report
Comments and Suggestions for Authors
Dear Authors,
This study investigates the effect of different mulching methods on increasing soil water storage capacity in the semi-arid climate conditions of the Loess Plateau. Experimental results indicate that SSM has a positive impact on increasing soil water storage capacity and improving ecological balance. PMP was found to increase soil temperature and catalase activity, although it did not decrease soil temperature. Additionally, mulching practices were observed to increase soil enzyme activities, which positively influenced plant productivity. The study demonstrates the effectiveness of mulching methods in increasing potato yield and improving WUE. Particularly, straw strip mulching plays a significant role in enhancing yield and promoting environmental sustainability.
I commend the authors for their significant contribution and commendable presentation. It has been a pleasure to engage with such a well-crafted manuscript, which stands out for its clarity and depth. However, upon careful examination, I have identified several overlooked aspects within the study, which I have annotated section by section in the attached manuscript file. It is imperative that these concerns be addressed, particularly within the Materials and Methods section, where clarity and precision are paramount. Furthermore, I recommend addressing any identified gaps within the Discussion and Results sections to enhance the comprehensiveness of the manuscript. Upon addressing these suggestions, I believe the manuscript will be well-positioned for acceptance.

Comments on the Quality of English LanguageMinor editing of English language required.
Author Response
Dear Reviewer:
Thank you for taking the time to review this manuscript. We really appreciate all your comments and I have benefited a lot from the comments on the manuscript, these comments would help to improve the quality of our manuscript. The following are the detailed corrections to our manuscript entitled “Mulching practices improve soil moisture and enzyme activity in drylands, increasing potato yield” (ID: agronomy - 2998306).
Comments 1: Please write the type of potato used and its properties. Response 1: Thank you for pointing this out. We agree with this comment. Therefore, we have increase type of potato used and its properties. [Tested material is “Longshu 7”, which belong to medium-late maturing variety]. In the revised manuscript this change can be found – page3, line 136. |
Comments 2: Also give information about planting distance and spacing. |
Response 2: Thank you for your comments. All the information about planting distance and spacing in the Materials and Methods (Figure 2) and we have explained. |
Comments 3: Please explain the methods in detail. How did you perform the analyses? How was catalase activity analyzed in soil? Response 3: We gratefully for your comments. We agree with this comment. Therefore, we have deleted the previous description of the method and added analytical methods. [Soil urease activity was determined by the sodium phenol-sodium hypochlorite colorimetric method and expressed as milligrams NH3-N in 1 g soil after 24 hours. Catalase activity was determined by potassium permanganate titration, expressed as milligrams of 0.02 mol/L potassium permanganate consumed by titration of 1 g of soil after 20 minutes. Sucrase activity was determined using the 3.5-dinitrosalicylic acid to calculate the quality of glucose produced in 1 g of dry soil after 24 hours]. In the revised manuscript this change can be found – page5, line 185-195. |
Comments4: deleting “Generally speaking” Response 4: Thank you for pointing this out. We agree with this comment. Therefore, we have deleted “Generally speaking”. In the revised manuscript this change can be found – page 6, line 253. |
Comments 5: Rewrite Line 270. |
Response 5: Thank you for pointing this out. We agree with this comment. Therefore, we have changed “but none of them were significantly different from CK in the 20–40 cm soil layer” to [No significantly different from CK in 20-40 cm soil layers]. In the revised manuscript this change can be found – page 8, line 292 - 293. Comments 6: I'm not sure of the correct unit for catalase activity, please clarify this. Response 6: Thank you for pointing this out. We agree with this comment. Therefore, we have change catalase activity unit for “mg·g·20·min”. Mention exactly where in the revised manuscript this change can be found – page10, line 311, (Figure 8). Comments 7: Please check the spellings. I think there are mistakes that you may have missed. Response 7: We were really apologized for our careless mistake. Thank you for your reminder. We have changed “7.6a” to “7.6ab”. In the revised manuscript this change can be found – page11, Table 1. Comments 8: Please include a paragraph discussion of correlations. Response 8: We gratefully for your comments. We agree with this comment. Therefore, we have used a paragraph discussion of correlations. [Hou et al [49] demonstrated that the effect of soil moisture on wheat yield showed a direct contribution. Wu et al [50] through the analysis of potato yield and water-temperature-fertilizer flux pathway found that soil moisture under different treatments had the greatest effect on potato yield. The study revealed that the soil moisture under various mulch treatments influenced the yield of potatoes. Temperature was identified as the next most influential factor, with the different mulching techniques enhancing soil enzyme activities and improving the hydro-thermal conditions of the soil. These enhancements ultimately resulted in a notable increase in potato yield.]. In the revised manuscript this change can be found – page 14, line 489 - 496.
we would like to express our great appreciation to you and reviewers for comments on our paper.
|